# The Impacts of Digital Technology on Service Design and Experience Innovation: Case Study of Taiwan's Cultural Heritage under the COVID-19 Pandemic

**Wai-Kit Ng** [1], **Fu-Tien Hsu** [2] **and Chun-Liang Chen** [1,*]

1    Graduate School of Creative Industry Design, National Taiwan University of Arts, New Taipei City 22058, Taiwan
2    Department of International Business, National Taiwan University, Taipei City 10617, Taiwan
*    Correspondence: jun@ntua.edu.tw

**Abstract:** The aim of this research is to identify the digital technology impact and experience innovation of cultural heritages in the context of the epidemic. The authors created an analytical framework and used a qualitative exploratory multi-case study of three cultural heritages in Taiwan. The findings indicate that digital technology has facilitated further innovations in cultural heritages under the epidemic to be closer to consumers' daily life and more connected with the young generation. Compared to traditional cultural heritages, profit-making cultural heritages need sales of its products to sustain operations, while live streaming, which is interactive, is rising as a new way to promote sales. Using multiple digital platforms can maintain consumers' interest in the cultural heritages, encouraging follow-up visits and thus resulting in more traffic online and offline. This paper illustrates the advantages of digital technology in the context of the epidemic, highlighting the innovative technology of live streaming and social platforms introduced that are different from the traditional cultural heritages.

**Keywords:** service design; digital technology; experience innovation; cultural heritage; COVID-19

## 1. Introduction

The majority of COVID-19 cases are caused by interactions in an indoor environment [1]. According to the data of the UN's cultural watchdog, approximately 85% of heritages have been temporarily closed due to the epidemic [2]. Especially, the use of digital tools in cultural heritages is receiving increasing attention [3]. However, cooperation between cultural heritage and technology is currently limited [4]. Research on cultural heritage management is mainly from the perspective of demand (visitor satisfaction) and the ability to protect cultural heritage [5]. Due to the demand for service improvements generated by COVID-19, much cultural heritage around the world still needs to supplement their strategic approach to digital transformation [6].

The global epidemic has a substantial social and economic impact on cultural heritage. The innovation of digital media has given the cultural heritage industry a new opportunity to continue interacting with visitors and creating value. This study adopts a multiple case study approach to extend the concept of experiential innovation to cultural heritage using service design as an entry point including the space service design and setting up digital platforms. This research surveyed three cultural heritages in Taiwan. These three properties represent revitalised and commercially viable cultural heritage. By the following research questions: (1) How do Taiwan's cultural heritages use digital platforms to interact with consumers? (2) How can the service design change the customer experience? (3) How does digital technology impact innovation on visitors' experiences during COVID-19?

The number of international tourists visiting Asian countries has increased annually. For example, the percentage of international tourists visiting Asian countries increased from

approximately 15.5% in 1995 to 24.9% in 2016 [7]. Specifically, according to statistics from the Tourism Bureau of the Ministry of Transportation, the number of tourists in Taiwan exceeded 10 million for the first time in 2016, and the trend of tourist growth is expected to continue [8,9]. The Taiwanese government also noticed the crucial role of tourism expansion in economic development and is eager to promote tourism internationally [10].

## 2. Theoretical Background

This section reviews the literature in service design, digital platforms, and experience innovation theory. We try to explain the relationship between service design and customer experience in cultural heritage and how digital technology can act as an enhancement to them to solve the plight of cultural heritage. Then, the third part of this section explains the research model, which presents the impact of digital technology during COVID-19 on cultural heritages.

The impact COVID-19 has had on the service industry is major, and transformative research is needed to address the sustainability and benefits of the customers, employees, and service organizations in travel-related industries [11]. The asymptomatic cases in Taiwan made the low infection rate a thing of the past, and it remains unknown how these cases occurred and how many more cases there are [12]. Many precautions and prompt responses have been taken against the virus in Taiwan, such as wearing masks and avoiding mass gatherings [13]. Digital technology, while used to identify the epidemic and its transmission, serves as a temporary alternative to the "physical entity", facilitating greater social participation in the cultural heritages and playing an even more significant role in the future [14]. It allows consumers to interact with brands in new and unconventional spaces through sensory experiences, gaming platforms, or mobile apps [15], and online shopping platforms with technological features provide users better shopping experiences [16]. Based on this, the following discussion will focus on three theoretical bases: service design, digital platforms, and experience innovation.

### 2.1. Service Design

Service is the basis of creating value, and service design is an exploratory process to build new value relationships among participants [17]. As service innovation brings new service ideas to life, developing a new prospect through new service design methods and models for the design is an important direction for service design research [18]. In addition to the cultural products themselves, cultural managers need to understand that an appropriate service design helps promote the participants' experience and post-consumption behaviour [19]. Service design can guide multiple stakeholders to be involved in platforms and share resources to enhance the value creation [20]. Further explanations of how and when service design is used will help service designers and managers to understand what environments are needed to foster innovation [21].

Integrating local art and culture into sustainable service design can create unique value and experiences for customers. As a means of service design, local value creation creates sustainable added value for local stakeholders [22]. Service design originates from a generation of digital-based network thinking. The customer is transformed into exquisite consumption, and the pursuit of the quality of life, innovative use of technology networks, and people will bring better service quality [23]. Based on this broad vision, the experience can be evoked through products, packaging, communications, store interactions, sales relationships, and events [24]. Technology applications will improve the experience dimensions and engage consumers to actively be a part of and create together the service experience [25]. The platform information will effectively improve the operation efficiency, help adopt customized and flexible services, and boost the service [26]. Online retailers need to pay attention to the customers' shopping experience process, as each factor can influence customer attitudes and willingness to purchase again [27].

Heritage faces the direct impact of the workforce and finances. Their inability to operate forces them to close, postpone, or cancel their services to their audiences. In order

to maintain their functions and increase access to cultural heritage, cultural institutions around the world have turned to digital platforms [28]. The social platform pages can interact with customers and generate income directly in the online market. Therefore, management must understand how to design a valuable interactive experience for a new interactive space [29].

### 2.2. Digital Platforms

With the help of digital technology, the platform provides an environment to link consumers' creative interactions with the experiential outcomes of potential resource capabilities [30]. Advances in digital technology have expanded the horizons of cultural heritage learning and provided new opportunities for information sharing through digital forms [31]. Technology has become the primary means of cultural heritage communication. Such sites that cannot operate offline continue to work online [32]. Managers of these sites recognized the need to better connect objects to the audiences' personal daily lives. Social media and e-commerce, for example, are used to attract and retain customers, especially millennials and the tech-savvy [33].

The innovation of service providers, namely striving to be customer-, technology-, and cocreation-oriented, determines satisfaction and eventually affects customer loyalty for intentions of referrals and recontracts [34]. Technology has the advantage and importance of creating an immersive and positive experience and behavioural purposes [35]. Online connection grants customers access to efficient, convenient, and real-time value. The establishment of social platforms enhances brand competitiveness and brings consumers opportunities to experience products and services [36]. The design of the tourist experience should coordinate different elements of integrating the digital world and the physical space into a unified and comprehensive experience [37].

### 2.3. Experience Innovation

In response to these market trends, cultural heritages are working to develop experience-based value creation activities that allow visitors to participate in the daily exhibitions and be a part of them [38]. The tourism experience is closely linked to the visitor's perception of the activity and the environment. Social media is a way for them to share a memorable experience with others [39]. Cultural heritage now requires creativity and innovation to develop intangible resources and new visitor experiences [40]. A key aspect of being consumer-oriented is providing added value by offering novel and meaningful services, in which technological innovation undoubtedly plays a vital role [41].

Service innovation and customer satisfaction are essential components of a company's sustainable development strategy [42]. To achieve sustainable development, companies need to provide innovative services constantly. Service design can be used to envision and design new services to achieve change [43]. Technology advances are pivotal to the digital transformation of services and understanding of the digital and data marketplace [44]. Visitors shape the service experience during participation, and technology facilitates new interactions, including on-site and online interactions [45]. The use of technology will improve the experience dimension and actively engage consumers to create the service experience. Such active participation requires experience in management planning and activities initiated for customers [24]. It needs to be further proven how institutions can rely on various digital tools such as live streaming and other social media platforms to stay connected and always be present for audiences who live in different communities around the world [46]. Digitization has become part of the museum's mission, structure, and practice, while the entity and digitization can be practised together as an enhancement of one with the other [47].

## 3. Methodology

*3.1. Case Studies*

This study is a qualitative study and will adopt a grounded theory and case study in qualitative research. Through multiple case studies, samples are designed to compare the service design and customer experience of three cultural heritages. The analysis process includes open coding, axial coding, and selective coding [48]. The existing literature recognises that digital platforms can create value for cultural heritage innovation, but it has not been linked to customer experiences, and further research is needed. Therefore, the theory has to be established to prove that the strategy in the research is synchronised with the new phenomenon [49]. Multiple case studies allow the researcher to look into the cases and observe differences between them [50]. In particular, the question of why and how phenomena such as the current plight of cultural heritage are being addressed [51]. Case selection is based on six basic types proposed by Langrish [52]: (1) Comparative: They are Japanese buildings of the same period that have been reused in different forms and contain different scales. (2) Representativeness: There are still in business under COVID-19 and have a creative department. The main consumer groups are tourists. (3) The best practices: All three cultural heritages have won government or NGO/industrial association awards. (4) The ones next door: the researchers are close to the research objects, with access to the survey of the management sector and consumers without the epidemic influence. (5) The "car, look at that": few cultural heritages are commissioned for third-party operations, especially commercial ones. (6) The economic: All three heritages have similar structures (competitors, buyers, etc.) that will allow for generic insights. In the case study, the researcher can collect and conclude from the survey data to understand the phenomenon in the study [53].

*3.2. Case Selection*

This study uses "intentional sampling" for the selection of interviewees. The data sources include primary and secondary data. The primary data was collected through semi-structured interviews with a total of 11 stakeholders from the management, staff and vendors of the cultural heritage site, experts, academics, and visitors during June and November 2021, which were recorded with permission (Table 1). The interviews covered the heritage operation status, innovation experience activities, the digital platform establishment, and the epidemic's impact (Appendix A). In addition, secondary data from various sources were collected to supplement this study, including interviews in magazines and newspapers, company websites and news and annual reports, government reports such as press releases from the Ministry of Health and Welfare, and reports on the operation of cultural heritages for a supplement.

**Table 1.** Data sources.

| Data Type | Case | Position | Type | Date |
|---|---|---|---|---|
| Interviews | The Red House | Director | phone interview | 2021/6/28 |
| Interviews | The Red House | Merchant full-time staff | Face-to-face interview | 2021/11/6 |
| Interviews | Hayashi | Head and Deputy Head of Marketing Planning Department | Face-to-face interview | 2021/5/6 |
| Interviews | Jin Jin Ding | full-time staff | Face-to-face interview | 2021/10/6 |
| Interviews | Scholars And Tourists | Three Taiwanese residents | Phone interview | 2021/5/17 |
| Interviews | Tourists | One Taiwanese residents | Face-to-face interview | 2021/11/5 |
| Interviews | Tourists | One Foreign tourists | Face-to-face interview | 2021/11/5 |
| Direct observations | Jin Jin Ding The Red House Hayashi | | Field Research | 2021/10/6 2021/11/5 2022/4/28 |

Sources: complied with this research.

### 3.3. Data Collection and Analysis

This study used triangulation on the semi-structured interviews with consumers and the management, secondary sources, and participant observation. The interview is the primary data source for qualitative research. At the same time, participant observation is commonly used in the case study, with in-depth descriptions and analyses of specific phenomena or phenomenon collections to achieve theoretical saturation during the triangulation [54]. By examining the same conclusion reached by different methods or from different observers, the research outcome can be verified [55]. During the period, methodological flaws or data or researcher bias were identified and eliminated [56]. This research collected comparable data materials about relevant phenomena for classification and coding based on grounded theory. The concepts and scopes were extracted and connected to form an approach based on these data [57], as specified in Table 2.

**Table 2.** Material categories.

| Category Code | Second-Order Code | Open Code |
|---|---|---|
| Service design | The change of service procedures<br>The involvement of other participants<br>New resources | Physical operation turned online and changes of operation time and items; new resources (epidemic prevention equipment)<br>Exterior partners (tourism program, restaurants and diners), food delivery platforms (food panda and uber eat); new products, epidemic prevention equipment (temperature measure and alcohol machine), and new raw materials |
| Digital technology application | New technology introduced<br>cooperation function<br>API | Appointment system, message real-name system QR code, and connection to partner commerce platform; use of digit payment, mobile payment, and digital coupons |
| Experience innovation | New shopping method<br>Dynamic interaction | Live streaming shopping discounts; more interactions on social platforms |

Sources: complied with this research.

## 4. Case Briefing

### 4.1. HAYASHI

HAYASHI is the only remaining department store in Taiwan and the only department store in Taiwan with a shrine. It is positioned as a cultural and creative department store. Before the epidemic, its monthly visitor flow reached up to 200,000. It won the Design for Asia Awards in 2016 and is currently the most popular must-see attraction and a new landmark in Tainan City. In 2016, it started to operate an online shopping website and entered Pinkoi in 2019. The physical site has been closed since 19 May, and it began to sell products by living streaming on Facebook on 1 June. The digital technology is mainly used in online shopping stores and social platforms, the previous built-in cooperation now changing to be independently run for membership.

### 4.2. The Red House

It is the only remaining public market building from the Japanese rule period in Taipei, located in Hsimenting. It was later positioned as a cultural and creative colony, hoping that it would be a living cultural and creative place to introduce young Taiwanese cultural and creative brands to more young people and tourists from abroad. In 2019, it welcomed monthly visits from 800,000 people, but the number dropped to 200,000–300,000 in 2020 after the epidemic set off and was ranked the fourth among the major tourist sites in Taipei in 2017–2021. The Red House, an international cultural and tourist landmark, won the 7th Taipei Urban Landscape Awards to revitalize historic spaces. It was shut down on 14 May and currently operates on social platforms with digital technology.

### 4.3. Jin Jin Ding

Jin Jin Ding is situated on 84 and 86 Jinhua Street tucked inside the Nishiki-Cho Japanese-style dormitories, which previously served as the staff dorm for the execution ground. Built between 1920 and 1930, it has kept the original Japanese-style structure, went through five years of renovation, and reopened to visitors two years ago. It has souvenir and cafe areas with products aimed at middle- to high-end consumers. The main visitor sources are tourists from Japan, Korea, and Mainland China and revenue sources from souvenir sales. Since the epidemic, sales mainly come from residents during festive seasons. Its current digital technology application is the online store and social platforms, independently run by Jin Jin Ding.

Table 3 summarizes the characteristics of the case cultural heritages, including their location, year of establishment, visitor traffic, received award, main customers, etc.

**Table 3.** Brief profiles of the case companies.

| Characteristic | Company | | |
| --- | --- | --- | --- |
| | Hayashi | The Red House | Jin Jin Ding |
| Prosperity | 1940 | 1940–1950 | 1930 |
| Location | Tainan City | Taipei City | Taipei City |
| Original purpose | Department Store | food market-Theatrical Theatre | Government Officers' Quarters |
| Local Cultural Characteristics | Japanese period architecture | Japanese period architecture | Japanese period architecture |
| Visitor traffic before the COVID-19 | 100,000–200,000 | 200,000–300,000 | 20,000–30,000 |
| Award | Design for Asia Awards | Urban Landscape Awards for historical space revitalization | Historical space revitalization Awards of Taipei City |
| Digital platform operation | Cooperative | Cooperative | Independent |
| Main customers | Foreign tourists especially Japanese | Tourists from Japan, Korea and Hong Kong t and local homosexual groups | Tourists from Chinese Mainland and Japan, Taiwan locals |

Sources: complied with this research.

## 5. Research Findings

### 5.1. Digital Technology and Experience Innovation

5.1.1. Social Platforms Help Cultural Heritages to Maintain Customer Relationship

The use of digital technology has helped Heritage to maintain its relationship with its customers. The management explained the following to us during the interview:

> *"We have been promoting various issues about the hayashi department store, the brand and the product stories to give positive energy during the epidemic. This is why we are promoting different topics about the department store. The department store will promote different topics such as music, literature, art and literature."*

The management of HAYASHI department store explained that they would organise events and post about the department store on different social media platforms to enhance the relationship with customers. Digital technology has helped these cultural institutions open again, and their online activities have increased significantly [26]. Table 4 showed the case company social platform interaction comparison.

**Table 4.** Case company social platform interaction comparisons.

| | Company | | |
|---|---|---|---|
| | **Hayashi** | **The Red House** | **Jin Jin ding** |
| Social media | FB, IG, Line | FB, IG | FB, IG, Line |
| Upload frequency before | every two days | every two days | every week |
| Upload frequency after | Daily | Daily | Three to seven days |
| Content before the COVID-19 | Announcement, publicity, and reply | Announcement, publicity, and reply | Product operation time |
| New in the COVID-19 | live streaming | - | live streaming |
| Function | Shopping, group buying, membership | - | membership |

Sources: complied with this research.

5.1.2. Digital Technology Helps Cultural Heritages Switch Services Quickly after a Shutdown

Cultural heritages have been building digital platforms for a long time, and their development accelerated during the epidemic. Tainan tourism authorities have proposed in their political guidelines to build a digital brand marketing platform and instruct the digital transformation of the tourism industry. A HAYASHI manager said to us:

> *"Since last year's epidemic, HAYASHI has been working hard to develop online shopping, so this year, we go with live streaming and social group management and promotion, which is relatively helpful during the epidemic."*

The Red House has adopted a strategy of "content management" and "in-depth marketing", starting with a Facebook fan page and later opening an Instagram account. As one of the informants mentioned:

> *"All the art and cultural performances have been canceled, but we will use Facebook and Instagram to help the art and cultural units repost, and then introduce the fair and workshop brands."*

HAYASHI uses the most digital platforms among the three cultural heritages, with an official online shopping store, collaborative platforms, and mobile app. The more innovative companies are more capable of using digital platforms to be more agile [58].

5.1.3. Cooperation with Online Shopping Platforms Benefits Cultural Heritages Operation

Cultural heritage needs to work more closely with the other partners to help innovate online shopping platforms. Cultural heritages that cannot build their own online shopping platforms will face more severe financial problems than others during the epidemic. Here are some responses from our interviewees:

> *"For the Red House itself, we cannot make our own online store; most of our brands run independently. We cannot make an online store for something we did not develop."*

Both The Red House and HAYASHI are stores that house brands, but HAYASHI is comprehensive and includes products that it creates. It reaches more foreign customers through platform cooperation and can maintain interactions with Lin's main consumers and tourists. One informant mentioned:

> *"We have been responding to foreign customers through Pinkoi, and we have launched on the Japanese market of Rakuten."*

Joining new platforms will open up more potential opportunities, but it will also prevent the cultural heritages from operating independently.

> *"We don't own the online platform, so we don't have the customer information. That's why we want to develop our membership system quickly."*

*"We hope to obtain more member data and consumption preferences to help HAYASHI's precise marketing in product development and event planning."*

Jin Jin Ding joined the MOMO e-commerce and Uber Eats during the epidemic and did improve the plight that they could not operate physically. Meanwhile, the collaboration with another cultural institution, Taipei Palace Museum, also brought new jobs for staff on unpaid leave.

With restriction removal and market reopening, there may be a fundamental shift in technology adoption and demand planning, which will let employees take on new front-line roles. Small and medium enterprises can achieve internationalisation goals and work with outside partners through digital platforms, which is an effective means for them to overcome challenges related to knowledge or resource constraints [59,60]. Companies should develop strong relationship networks, select quality partners, and link them to their core R&D teams to develop an integrated platform [61]. New functions added to existing online channels may bring new customers to talk about the unmet needs and thus lead to new sales opportunities [62].

In sum, Table 5 shows the comparison of case company technology adoptions.

**Table 5.** Comparison of company technology adoptions.

|  | Hayashi | The Red House | Jin Jin ding |
|---|---|---|---|
| Post | ✓ | ✓ | ✓ |
| Live steaming | ✓ | - | ✓ |
| Digital membership | ✓ | - | ✓ |
| Self-employed online store | ✓ | - | ✓ |
| Cooperation online store | ✓ | ✓ | ✓ |
| API | ✓ | ✓ | ✓ |

Sources: complied with this research.

*5.2. Service Design and Experience Innovation*

5.2.1. Technology Updates Benefit Service Improvement

During the epidemic, Jin Jin Ding updated its online store, as the original website was slow, difficult to create orders or find an item, and needed staff assistance for purchases. The brick-and-mortar stores used to be the main channel for customers to buy from, but after the new website was launched with the ordering problems improved, customers switched to online ordering, which made the latter the preferred alternative shopping form during the epidemic. Online shopping is convenient and simple and provides more choices than traditional stores [63].

Cultural heritage actively uses digital technology to innovate the customer experience, particularly linked to festivals and local communities. For example, the Red House used AR technology to organise parent–child events "catch ghosts" in the surrounding community on Halloween. Such events enhance the relationship with the community and have the effect of enlivening the surrounding economy. In terms of outbreak management, in the holiday creative market (regular events during the Red House weekends), suppliers and visitors will need to provide proof of vaccination, and alcohol will be available at the show desk for disinfection. The HAYASHI online shopping network has added brands and products such as fragrances, accessories, and refined daily wear that were previously only available in brick-and-mortar stores. Before entry, precautions like temperature taking at the forehead, hand disinfection, and code identification are required.

5.2.2. Application Programming Interface Connection

The Red House mainly uses its official website and the brand's independent stores connected to social media platforms for online sales. The operator told us that "people see the introduction on our Facebook and turn to their shopping store to purchase". Each

cultural heritage has a variety of payment methods. Jin Jin Ding accepts five kinds of digital and mobile payment platforms, and the store staff said that customers use mainly mobile payment such as Apple Pay and credit cards. The Red House uses mainly Line Pay and EasyCard, along with the digital coupon issued by the government. HAYASHI uses a dozen mobile payments such as Apple Pay, Samsung Pay, Google Pay, Line Pay, other digital port payments, and Pi wallet. This month, it also included EasyCard, the first time the payment platform will work with the Tainan department store industry.

### 5.2.3. The Change of Service Procedure May Bring New Opportunities

The Red House adjusted its business hours during the epidemic. On weekdays, it offers self-service, allowing customers to pay for a drink and have a place to rest and chat, cutting the need for the staff and better-attracting consumers who are affected by the prices. A store owner inside said that the business was not good since the shutdown, but they changed their operation to weekends and the Red House fair and holiday crowds, which helped product sales. During the epidemic, Jin Jin Ding took the initiative to provide customer services such as phone orders and delivery, which increased customers' willingness to purchase again. As for the products, they improved according to customer feedback, such as making package sales into certain combinations for them to choose from, and some products are sold separately. The packaging is up to the customers to select or customise. HAYASHI provides ordering services through calls and Facebook, and customers can choose to pick up the products at the store or have them delivered. Companies need to pay attention to customer relationship management to improve the ability to perceive the customers' asset level in the markets [64].

### 5.2.4. The Membership System Helps to Understand the Consumer Needs with Zero-Contact

The online store cannot be in physical contact with the consumers, and the latter's shopping references and purchase records help with the marketing of cultural heritages. Such heritages in Taiwan have recently established a membership system on the Line@ social platform. Before the epidemic, Jin Jin Ding mainly depended on Line, e-mail, and phone to inform the old customers of the latest products. After the new official website was launched, it set up its new membership system there. Jin Jin Ding also said that the membership system allows them to promote and let customers purchase without being in the store.

The managing sector of HAYASHI said:

> *"This year, we worked on line@ membership and hoped that we can learn about the customers' purchase information and motive through the backstage data."*

Information on user experiences serves to assess the service quality and expectations [65]. The alignment between the consumers' personal preferences and the brand image will affect how they perceive the brand assets [66].

### 5.3. Service Design and Digital Technology
### 5.3.1. Live Streaming Is a New Trend in Cultural Heritage

Live streaming brings up a new way to interact with cultural heritage. During the epidemic, cultural heritages use social platforms to sell their merchandise, but their customer crowds still hesitate to embrace this way of promotion. The Red House 108 annual report mentioned adopting a visual way of communication to cater to the young generation and seek greater publicity through online media and live streaming. According to HAYASHI's feedback, they adopted such a practice for the first time with limited coverage, so the sales volume did not hit a peak right away, but the interactions with followers and their recognition of the brand turned out to be pretty good. Different from other department stores' sales, Jin Jin Ding has more means to communicate with the customers, and they also leave personal contacts from Line or e-mail. One informant said:

> *"Since the beginning of the epidemic last year, hayashi department store has been working hard on online shopping. Therefore, this year, with live streaming and social media operation and promotion, it is relatively good to help in the epidemic."*

Live streaming marketing can help companies to maintain business or even growth during the epidemic of COVID-19 [67]. However, the effects may be affected by the customer group.

### 5.3.2. Discount Activities Design Helps Improve Customer Experience

The three cultural heritages have all launched events such as set sales, discounts, coupons for membership registration, and exclusive discounts on certain platforms, which helps to attract customers to the online shopping platforms. HAYASHI also had many discount events in their online and offline stores.

> *"HAYASH shopping website has set up specially discount service for anti-epidemic area. At the end of May there was group purchase in limited time for people in the middle and north part who missed Tainan to bring these fine products home at the discounted price."*

Jin Jin Ding presents different coupons and discounts to different members according to their previous consumption amounts. It is also building itself on different shopping platforms with events to attract customers. With digital and paper discounts issued by the Taiwan local government, stores in the Red House also issued different coupons to boost sales. One of the respondents said that it was their biggest discount ever.

> *"Now the products we sell are 50% off for the second item they buy, the biggest discount ever and the long-shirts are the new products and they don't have this discount."*

According to the consumers' common responses, the cultural innovation products are mostly sold at relatively high prices, and they would buy there once at most. Promotions help customers to accept the prices, and the improved shopping experiences and satisfaction will make them more willing to revisit [68].

### 5.3.3. Content Promotion and Online Events Help with Network Traffic

During the epidemic, staff at the Jin Jin Ding brick-and-mortar stores explained and promoted the online ordering to the customers that visited, which led to a surge in online sales during the Mid-Autumn Festival. The other two cultural heritages preferred activity on social websites. HAYASHI changed its paper DM to an online one on social platforms to promote its products. There are also discounts that need interactions with people, for example, putting the store number on Line@ to obtain a "HAYASHI exclusive discount". These limited digital coupons are issued every weekend. Those who own a five-time coupon with the HAYASHI official Line account get an extra discount. Log in to HAYASHI for a consumption invoice, and there will be a lottery for hotel accommodations and dining coupons that are worth over USD 10,000. A respondent said that the consumers took active part is this. Online interactions allow consumers to constantly connect with the cultural heritages, and the service design should include meeting their emotional needs (including artistic events) [69]. Here are some quotes from the informant.

> *"Since the epidemic HAYASI started to spontaneously promote on social websites its daily details, brand and product stories for, so as to tell people about their physical events like music, literature and art for them to look forward to one day they can visit again when the shutdown is removed and there is no safety concern."*

The findings above responded to the positive influence [24,25] brought about by experienced innovation. The digital technology interaction model presented in this research (Figure 1) can be used to explain how cultural heritages use digital technology to connect with consumers during this epidemic. Their management is affected by the operation; the service design, procedure improvement, and experience innovation can maintain or establish a connection with consumers. The use of digital technology allows both parties to interact on social platforms and online shopping stores when they wish to, allowing

consumers who are unable to leave their homes to gain an experience and connect emotion-ally with these cultural heritages. When physical interactions are absent, the management sector and consumers continue to interact and create value to the social platforms and online store created by digital technology. This model can also be applied to discuss cultural circuits Champ and Brooks [70]. Relevant research on cultural circuits have been complied, including areas of public relations, tourism, and so on. Those applied to management goals witness interpersonal relationships in a wide range of experiences, including the process of representation, identity, production, consumption, and regulation. This model of a cultural circuit can apply as long as the processes of understanding the cultural circuit are dynamic and see how, when, and why they emerge; how they are related; and when the archetypes change [71]. Put our model in the cultural circuit, and we can identify the five aspects of performance, identity, production, consumption, and regulation [72]. The influencing process is dynamic and is involved with the manager's experience innovation. The epidemic has influenced the process and enhanced the speed of technology applications and the entity limitations.

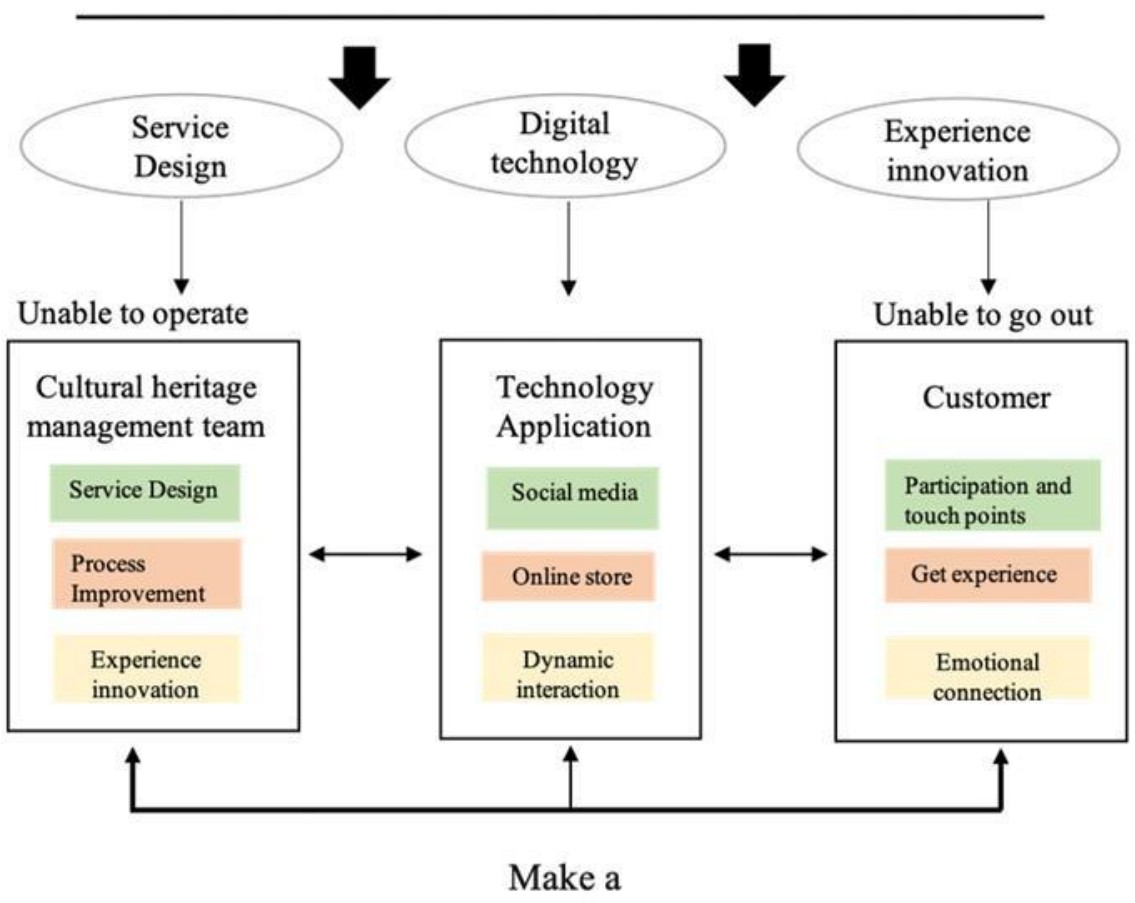

**Figure 1.** Digital technology interaction model.

## 6. Discussions

During the COVID-19 epidemic, we have integrated virtual and reality customer experiences through service design, which is conducive to the practice of heritage cultural innovation. When there is a creative department, there are connected places local artists and craftsmen have the opportunity to supply, which can bring the power of stabilising society. Consider the importance of local traditions and culture in service development to increase profits while maintaining local cultural values [73], and shift the focus of art

from elegant culture and historical value to materials and commodities that can be touched in daily life. The role of traditional art threatens the economy, but now, it also benefits from the renewal of the industrial structure. Readjust the relationship between art and society, become a new cultural intermediary for consumers and producers, go deep into their personal daily life, and connect the related industries [74].

Different digital platforms can bring about other benefits to cultural heritage. Facebook and Instagram, for example, can interact with customers through texts and posts, while Line can establish a convenient and simple membership system. Cultural heritages can run their own online store to obtain their membership data, and cooperation with other online retailers helps to expand customer sources and establish their own platform in the preliminary stage. Increasing the inspiration and participation caused by interactions is conducive to enhancing the interest in new knowledge and attracting the younger generation and potential customers [75].

This research responds to its multilateral platform function [20,30]), and here are the two propositions:

**Proposition 1:** *Cooperation with multiple platforms helps cultural heritages create new chances.*

**Proposition 2:** *Marketing events on different platforms will improve customer engagement.*

After the transformation of heritage, the atmosphere and service design of the whole field allows consumers to experience historical and cultural value by buying these cultural and creative commodities for experience and innovation. They can continue to feel historical value and educational significance after returning home. The way of industrial, artistic creation can further establish contact between heritage and tourists, interact through collective memory, and provide tourists with unique opportunities to experience pleasure and feel the effects of psychological well-being [76,77]. The results above echo the advantages of the proposed service design. Based on this, we have two propositions:

**Proposition 3:** *Optimising the service procedure and products in the product design helps cultural heritages to build emotional value during the epidemic.*

**Proposition 4:** *Analysis of the membership data helps the cultural heritage management team to have closer ties with the consumers.*

Service providers and consumers have shown passion and doubt about the use of technology, which might be negative in customers' experiences [63]. Klaus [78] concluded that the lack of interactions is the reason why some customers are not willing to buy online. The interpersonal communications and fine relationships will affect middle- to high-end customers' choices. It has been proven that the information technology of living streaming has a positive influence on customers' purchase decisions [79]. By emphasising the process and relationship of value creation, it is conducive to the coexistence of non-profit, commercial, and social enterprises in the organization and the realisation of the objectives and vision of cultural and art organisations [80]. Here are the propositions for the article:

**Proposition 5:** *The effect of experience innovation is influenced by customer engagement and where they come into contact.*

**Proposition 6:** *Cultural innovation products sold on a live streaming platform triggers new experiences for customers.*

## 7. Conclusions and Suggestions

*7.1. Conclusions*

This paper introduces the strategies and measures adopted by Taiwan's cultural heritage during the COVID-19 crisis, with particular reference to the use of technology and the establishment of digital platforms. The closure of the museum from 2020 to May 2021 has created a widespread damaging effect on cultural heritage. Cultural heritage has created new market opportunities during the epidemic through the ability to stay engaged with consumers through digital technology and through online shopping. Three main findings can be concluded: (1) The epidemic has facilitated cultural heritage's further innovation to be closer to the consumers' daily life of consumers and connect more with the younger generation. In order to adapt to the changing epidemic environment, digital technology has provided cultural heritages with a new channel to get in touch with consumers. (2) The use of digital technology helps to improve the quality of service and serves as a catalyst for innovation during the experience. Live streaming is a new trend in cultural heritage. (3) Online shopping stores help them to continue to serve the consumers, and cooperation with multiple platforms helps develop more opportunities. The online operation and the physical shops create a two-way diversion effect.

*7.2. Theoretical Contribution*

Based on the service design point of view, the research has observed a change from physical to online services, enhanced the relevant theory for the cultural heritage industry, and provided cases of measures taken against the COVID-19 restrictions.

(1) The research has identified the relation between the service design and consumer experience, especially under the COVID-19 impact. To be specific, the research also confirms that, during the epidemic, cultural heritages can use digital technology to interact with customers and build new chances in the market through online shopping stores.

(2) The research has stressed digital technology's advantage during the epidemic, especially the innovative introduction of live streaming and social platforms compared to traditional cultural heritages. This is in line with the current e-commerce development and has broadened the literature scope.

(3) Based on the above, the research has constructed a concept framework. It indicated that service design, experience innovation, and digital platform can be involved in the consumer interaction process to highlight the experience innovation influence and connect with customers. Digital technology brings new services and products, which are constantly creating value and being delivered to consumers. Additionally, the research also revealed the influence of the membership system. The cultural heritages change situations where they cannot be in direct contact with customers to understand or provide the proper services. The current discussion focused on the cost of membership and not its function in maintaining relationships with consumers.

*7.3. Practical Implications*

This research helps the cultural heritage industry learn about the transforming path and innovation strategy against the epidemic and sort out the ways of online operation.

(1) The global epidemic has lasted until today and is full of uncertainties. It has become a significant trend for companies to turn digital, and it is expected that people's shopping habits are changing fundamentally to become online. Brick-and-mortar stores need to think about introducing digital technology to interact with consumers for them to stay interested.

(2) Different social platforms do not have the same customer crowds, but companies can choose how to interact accordingly. Additionally, the managing sector should consider services and products related to the epidemic. It would inspire the company to renovate, as well as boost sales. Cultural heritages need a series of measures to help

consumers to confront the changes brought about by the epidemic (cancellation of courses and performances and changes to or a refund of online purchases). This is one thing service design should think about during an epidemic.

(3)    Cultural heritages revitalised need to review how to combine their cultural resources with daily life and recreate opportunities to continue interactions with customers. This paper has offered new references for the transforming industries, with more opinions on maintaining a connection with consumers [81,82].

### 7.4. Future Research Directions

The COVID-19 situation may go up or down, but there might be new dilemmas for the cultural heritages, for example, the lack of products or delivery staff. The managing sector should be more flexible and responsive in innovating against epidemic changes. At the same time, the spread of COVID-19 has limited our contact with the outside world to online, and there will be competition among the cultural heritages. It requires further discussion about staying competitive in the area. Live streaming is a new way of shopping and is still in the preliminary stages in terms of cultural heritage. This is now mainly popular in Asia and especially in China. Future research can pay more attention to this respect.

**Author Contributions:** Conceptualization, W.-K.N., F.-T.H. and C.-L.C.; Data curation, W.-K.N. and C.-L.C.; Methodology, W.-K.N. and C.-L.C.; Project administration, W.-K.N., F.-T.H. and C.-L.C.; Resources, F.-T.H.; Validation, F.-T.H.; Writing review and editing, W.-K.N., F.-T.H., C.-L.C.; Writing—original draft, W.-K.N.; Funding acquisition, Investigation and Supervision, W.-K.N. and F.-T.H.; and Formal analysis, W.-K.N. All authors have read and agreed to the published version of the manuscript.

**Funding:** This research received no external funding.

**Informed Consent Statement:** Informed consent was obtained from all subjects involved in the study.

**Data Availability Statement:** Not applicable.

**Conflicts of Interest:** The authors declare no conflict of interest.

## Appendix A  Interview Outline

For the management teams:

(1)    What kind of culture and spirit do you wish to show the consumers?
(2)    What's the main age and ethnic group are the consumers? Which country are they mainly from? Is there ethnic consumption difference online and offline?
(3)    What measures do you take to confront the epidemic? What difficulties do you have during the epidemic?
(4)    As the old houses cannot receive visitors at the moment, what measures have been taken for consumers to understand local culture and brand history?
(5)    How many social platforms do you run on now? How do these platforms help you communicate with consumer interaction?
(6)    What are the current projects available on the online shopping store? Is such long-term operation functional against the epidemic impact?
(7)    How do you present cultural innovation on such delivery products?
(8)    Do you run your store on the online store? Does membership help? What discounts are available for the members?
(9)    What's the impact of the new round of epidemic on Taiwan? How is it compared to last year?
(10)  What new measures were adopted after the shutdown?
(11)  I see that you have live streaming. How do customers respond to that? Does it promote product sales?
(12)  How do you maintain your connection with customers during an epidemic?

For Consumer:

(1) What do you think is the most prominent feature?
(2) What do you expect to feel in the environment, for example, experience events, commodities, and ambience? Is the experience in line with your expectation?
(3) How do the staffs serve and present themselves?
(4) Is there any inconvenience in the environment?
(5) Do you visit more or less than before? Do you gain any new experiences or feelings? Have you used the online platform?

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
