# Peer review of "The Impacts of Digital Technology on Service Design and Experience Innovation: Case Study of Taiwan’s Cultural Heritage under the COVID-19 Pandemic"

_systems, doi:10.3390/systems10050184_

Round 1
Reviewer 1 Report
The manuscript illustrates the strategies and measures taken by cultural heritages in Taiwan during the COVID-19 crisis, especially the technology use and the digital plat form establishment. Through multiple case studies, grounded theory and semi-structured in-depth interviews, three main findings can be concluded: (1) The epidemic has facilitated cultural heritage’s further innovation to be closer to the consumers’ daily life of consumers and connect more with the younger generation. In order to adapt to the changing epidemic environment, digital technology has provided cultural heritages with a new channel to be in touch with consumers. (2) Compared to traditional cultural heritages, profit-making cultural heritages need the product sales to maintain operation and the interactive live streaming is becoming a new sales tool. (3) Online shopping stores help them to continue to serve for the consumers, and cooperation with multiple platforms helps develop more opportunities. Frequent interactions on social media platforms will help maintain consumers’ interest in cultural heritages, which boosts return to the sites and bring bilateral interests of traffic.
The research theme is helpful to tourism under COVID-19. The research method and data source are reliable. The results are identical to what we known. To improve the manuscript, one main limitation needs to be revised in Part 5 about research findings. In this part, it's better to analyze the mechanism of the findings from interview results.
Author Response
Dear Editors and Reviewers,
Thank you very much for giving the author the opportunity to further revise the paper, and thank you very much for your valuable suggestions on the paper, which greatly improved the quality of the paper. The authors have revised the paper one by one according to your suggestions, and hope to receive your approval. The following is a response to the comments of reviewers.
Responds to the reviewer’s comments:
Reviewer #1:
The manuscript illustrates the strategies and measures taken by cultural heritages in Taiwan during the COVID-19 crisis, especially the technology use and the digital platform establishment. Through multiple case studies, grounded theory and semi-structured in-depth interviews, three main findings can be concluded:
(1) The epidemic has facilitated cultural heritage’s further innovation to be closer to the consumers’ daily life of consumers and connect more with the younger generation. In order to adapt to the changing epidemic environment, digital technology has provided cultural heritages with a new channel to be in touch with consumers. (2)Compared to traditional cultural heritages, profit-making cultural heritages need the product sales to maintain operation and the interactive live streaming is becoming a new sales tool. (3)Online shopping stores help them to continue to serve for the consumers, and cooperation with multiple platforms helps develop more opportunities. Frequent interactions on social media platforms will help maintain consumers’ interest in cultural heritages, which boosts return to the sites and bring bilateral interests of traffic.
The research theme is helpful to tourism under COVID-19. The research method and data source are reliable. The results are identical to what we known. To improve the manuscript, one main limitation needs to be revised in Part 5 about research findings. In this part, it's better to analyze the mechanism of the findings from interview results.
AR:
Special thanks to you for your good comments. The findings in part 5 and conclusions have been supplemented and revised to more clear and added more evidence from the interviews to support our opinion. For example, L34-37 on page 7 “We have been promoting various issues about the hayashi department store, the brand and the product stories to give positive energy during the epidemic. This is why we are promoting different topics about the department store. The department store will promote different topics such as music, literature, art and literature.”this response to the first point of the conclusion “Social platforms help cultural heritage sites form closer relationships with consumers.”
In addition, the subheading of 5.3.1 is amended to “Live streaming is a new trend in cultural heritage” to better summarise the conclusions L13-L15, also mentioned in page 14 “The use of digital technology helps to improve the quality of service and serves as a catalyst for innovation in the experience. Live streaming is a new trend in cultural heritage.”
Thank you again for these comments. We are grateful for your advice and suggestions.
Reviewer 2 Report
The general context of the problem and general research topic is not clear in the introduction. For example, it is much better explained in L96-101.
In its current version the paper is virtually impossible to follow as it constitutes a continuum of unconnected sentences. There is virtually no difference between a regular stop and a jump of paragraph. There is not a flow of ideas
Let me indicate some examples
L31: “85% of these sites are now closed” substitute now by a date (so that the paper remains informative as time passes).
L 33-37. All this sentence do nt seem conected. Improve the flowing of the text
L39: “To fill the blank in the literatura” you have not clearly indicated the gap in the literatura that you aim to cover.
L57-60: This first sentence is not clear, improve it.
L73-178. All this sections are not clear, and are extremely difficult to read.
There is a continuous of unconnected sentences with a reference. But there is not a linked discourse. Authors basically jump from one idea to the other, and many of those ideas are not elaborated to clarify their relevance and how they are linked into the paper.
Improve the writing so it is easier to follow.
Also many of these sentences are not well written, so the meaning is not clear. Some examples:
L105-107 “With the help of digital technology, interactive platforms…” you use interactive 3 times in that sentence, so the meaning is not clear
L118: what is “company’s customer performance”?
L130-131; “The tourist experience is closely related to tourists' perception of emotionally and physically”… emotionally and physically what?
L171-172: “(5) The “car, look at that”: few cultural heritages are commissioned to third-party operations, especially commercial 172 ones” what do you mean?
L184-185: “Secondary sources include newspaper and magazine interviews, company websites and press releases, and final accounts. The secondary sources include interviews in magazines and newspapers,” this is just repeated…
Author Response
Dear Editors and Reviewers,
Thank you very much for giving the author the opportunity to further revise the paper, and thank you very much for your valuable suggestions on the paper, which greatly improved the quality of the paper. The authors have revised the paper one by one according to your suggestions, and hope to receive your approval. The following is a response to the comments of reviewers.
Responds to the reviewer’s comments:
AR:
Thank you for your valuable suggestions. We have corrected the sentences in the introduction and removed the incomprehensible content, Corrected parts are marked in colour. Also, we attend to the English native speakers to correct them to make our article easier to read.
- L31, page1: “85% of these sites are now closed” substitute now by a date (so that the paper remains informative as time passes).
AR: Thanks for this suggestion. We have revised to “According to the data of the UN’s cultural watchdog, approximately 85% of heritage have been temporarily closed due to epidemics.” We would like to illustrate the impact of the epidemic on the physical operation of cultural heritage.
- L 33-37. All this sentence do not seem conected. Improve the flowing of the text
AR: L33-37 was modified to make it clearer, near L30-L35 now, page1.
- L39: “To fill the blank in the literatura” you have not clearly indicated the gap in the literatura that you aim to cover.
AR: To make the purpose of this article clearer the text has been changed to “This study adopts a multiple case study approach to extend the concept of experiential innovation to cultural heritage using service design as an entry point including the space service design and setting up digital platforms.”, near L36-38, page1.
- L57-60: This first sentence is not clear, improve it.
AR: We have provided additional content on L49-51, to explain what we are trying to explain in this section of the literature review.
- L73-178. All this sections are not clear, and are extremely difficult to read. There is a continuous of unconnected sentences with a reference. But there is not a linked discourse. Authors basically jump from one idea to the other, and many of those ideas are not elaborated to clarify their relevance and how they are linked into the paper.
AR: Thanks for this suggestion. We have now addressed the connected sentences with a reference. L73-178 these sections were revised to a more readable and purposeful description. We have removed too many redundant words from these sentences.
- L105-107 “With the help of digital technology, interactive platforms…” you use interactive 3 times in that sentence, so the meaning is not clear.
AR: We have revised these sentences to make it clear.
- L118: what is “company’s customer performance”?
AR: We have revised the “company’s customer performance” to satisfaction.
- L130-131; “The tourist experience is closely related to tourists' perception of emotionally and physically”… emotionally and physically what?
AR: Emotionally and physically is explained about visitor's perception, we have corrected this sentence.
- L171-172: “(5) The “car, look at that”: few cultural heritages are commissioned to third-party operations, especially commercial 172 ones” what do you mean?
AR: We have revised this sentence to make it clear.
- L184-185: “Secondary sources include newspaper and magazine interviews, company websites and press releases, and final accounts. The secondary sources include interviews in magazines and newspapers,” this is just repeated.
AR: We have deleted the repeated sentences.
Thank you again for these comments. We are grateful for your advice and suggestions.

Round 2
Reviewer 2 Report
The authors have made a remarkable effort to improve the paper’s readability, and to clarify the research objective.
I have one minor comment to add, given that the journal reader might not have deep tourism knowledge of the specific location.
I suggest adding a paragraph contextualizing tourism in Asia and Taiwan at the end of the introduction. Then, add some references, and relevant data on the importance of tourism for Taiwan and the huge growth of tourism in Asia
Some references for tourism in Taiwan might be:
https://doi.org/10.1016/j.tourman.2005.05.011
https://doi.org/10.1080/10941665.2011.539394
https://doi.org/10.18488/journal.31.2017.41.12.16
some references for tourism in Asia to include are:
https://doi.org/10.1080/10941665.2017.1359192
https://doi.org/10.1080/10941665.2018.1524774
Author Response
Dear Editors and Reviewers,
Thank you very much again for giving the author the opportunity to further revise the paper, and thank you very much for your valuable suggestion on the paper, which greatly improved the quality of the paper. The following is the response to the comment of reviewer.
Reviewer’s comments:
The authors have made a remarkable effort to improve the paper’s readability, and to clarify the research objective.
I have one minor comment to add, given that the journal reader might not have deep tourism knowledge of the specific location.
I suggest adding a paragraph contextualizing tourism in Asia and Taiwan at the end of the introduction. Then, add some references, and relevant data on the importance of tourism for Taiwan and the huge growth of tourism in Asia
Some references for tourism in Taiwan might be:
https://doi.org/10.1016/j.tourman.2005.05.011
https://doi.org/10.1080/10941665.2011.539394
https://doi.org/10.18488/journal.31.2017.41.12.16
Some references for tourism in Asia to include are:
https://doi.org/10.1080/10941665.2017.1359192
https://doi.org/10.1080/10941665.2018.1524774
AR:
Thank you for your valuable suggestion. According to your suggestion, we have added one paragraph at the end of the introduction also cited the suggested references. Revised parts are marked in colour.
Thank you again for these comments. We are grateful for your advice and suggestions.